# Traumatic Childbirth and Birth-Related Post-Traumatic Stress Disorder in the Time of the COVID-19 Pandemic: A Prospective Cohort Study

**DOI:** 10.3390/ijerph192114246

**Published:** 2022-10-31

**Authors:** Lamyae Benzakour, Angèle Gayet-Ageron, Maria Jubin, Francesca Suardi, Chloé Pallud, Fanny-Blanche Lombard, Beatrice Quagliarini, Manuella Epiney

**Affiliations:** 1Department of Psychiatry, Geneva University Hospitals, University of Geneva, 1205 Geneva, Switzerland; 2Faculty of Medicine, University of Geneva, 1206 Geneva, Switzerland; 3Division of Clinical Epidemiology, Department of Health and Community Medicine, Geneva University Hospitals, University of Geneva, 1206 Geneva, Switzerland; 4Department of the Woman, the Child and the Teenager, Geneva University Hospitals, 1205 Geneva, Switzerland

**Keywords:** birth-related PTSD, traumatic childbirth, COVID-19

## Abstract

Background: Birth-related post-traumatic stress disorder occurs in 4.7% of mothers. No previous study focusing precisely on the stress factors related to the COVID-19 pandemic regarding this important public mental health issue has been conducted. However, the stress load brought about by the COVID-19 pandemic could have influenced this risk. Methods: We aimed to estimate the prevalence of traumatic childbirth and birth-related PTSD and to analyze the risk and protective factors involved, including the risk factors related to the COVID-19 pandemic. We conducted a prospective cohort study of women who delivered at the University Hospitals of Geneva between 25 January 2021 and 10 March 2022 with an assessment within 3 days of delivery and a clinical interview at one month post-partum. Results: Among the 254 participants included, 35 (21.1%, 95% CI: 15.1–28.1%) experienced a traumatic childbirth and 15 (9.1%, 95% CI: 5.2–14.6%) developed a birth-related PTSD at one month post-partum according to DSM-5. Known risk factors of birth-related PTSD such as antenatal depression, previous traumatic events, neonatal complications, peritraumatic distress and peritraumatic dissociation were confirmed. Among the factors related to COVID-19, only limited access to prenatal care increased the risk of birth-related PTSD. Conclusions: This study highlights the challenges of early mental health screening during the maternity stay when seeking to provide an early intervention and reduce the risk of developing birth-related PTSD. We found a modest influence of stress factors directly related to the COVID-19 pandemic on this risk.

## 1. Introduction

The coronavirus disease 2019 (COVID-19), which spread all around the world after the first case in Wuhan in November 2019, induced isolation, stressful conditions related to the fear of dying and the many bereavements and socio-economic consequences. Since the beginning of the COVID-19 pandemic, many research studies have highlighted its great impact on mental health in the general population, with increased rates of depression, anxiety and PTSD in comparison with the prevalence of these psychiatric disorders before the COVID-19 pandemic [1]. Several studies have highlighted the negative impacts of COVID-19 pandemic on the access to and quality of maternity health services, in particular in the early phase of the pandemic due to restrictions on partner and visitor attendance [2,3] or due to adaptive practices in maternal care with the use of telemedicine [4]. Women also could potentially be anxious towards a risk of contamination for themselves or their fetus, but also towards vaccination against SARS-CoV-2 and its potential side effects. Data on the impact of COVID-19 pandemic on mental health issues for women who delivered during this period need to be explored further. 

While between 20% and 48% of women report a traumatic experience associated with the delivery [5,6], the taboo that surrounds it maintains the idea that this phenomenon remains exceptional and that it is limited to situations of obstetrical complications. However, childbirth with no medical complications, whether obstetric or neonatal, can nevertheless also be experienced as traumatic and can lead to complications in the mental well-being of the mother, the father and the child. The authors of a recent meta-analysis suggested the use of the terminology of birth-related post-traumatic stress disorder (PTSD) to describe a PTSD resulting in a traumatic childbirth and concluded that 4.7% of mothers developed birth-related PTSD and 12.3% of mothers developed birth-related post-traumatic stress symptoms (PTSS) [7]. Actually, a traumatic childbirth does not necessarily give rise to the development of birth-related PTSD, which is the most serious clinical evolution and can be limited to birth-related PTSS, which can include post-traumatic stress symptoms that are not sufficient to validate the criteria for a PTSD diagnosis. 

Birth-related PTSD and birth-related PTSS are associated with deleterious effects on breastfeeding, which is more prematurely stopped [8], as well as impacting child health and development, mother–child attachment and the quality of attachment [9], with deleterious effects on the couple’s relationship [10]. They are highly co-morbid with post-partum depression (PPD) [11]. Despite these long-term consequences of birth-related PTSD, there is currently no routine prevention of PTSD implemented in most maternity services, unlike prevention strategies for PPD, which are most commonly developed. However, many predictors have been identified and confirmed by numerous robust studies [9,12,13,14]. According to a diathesis stress model of the aetiology of childbirth PTSD, risk factors are divided into three categories: pre-birth, peripartum and post-partum [9,12,13,14]. First, pre-birth vulnerability factors are involved, such as antenatal depression, fear of childbirth, poor health or complications in pregnancy, PTSD related to a previous traumatic event and counselling for pregnancy or birth, suggesting help being needed during pregnancy or birth [8,10,11,12]. Concerning birth, negative subjective birth experiences, having an operative birth (assisted vaginal or caesarean), a lack of support and dissociation appeared as the most important risk factors [9,12,13,14]. In high-risk samples of women that had pre-eclampsia or emergency caesarean sections (and not in other women), marital status, poor health or complications in pregnancy and negative emotions in birth influence the risk of birth-related PTSD [9]. Concerning the post-partum period, PPD, other comorbid symptoms, stress and poor coping are identified risk factors related to childbirth PTSD [15]. A recent meta-analysis highlighted controversial findings for risk factors underlying traumatic birth and birth-related PTSD, showing that the factors involved in traumatic childbirth are different from those involved in birth-related PTSD [16].

A traumatic childbirth must validate the definition of a traumatic event according to the Diagnostic and Statistical Manual of Mental Disorders (DSM-5) to diagnose birth-related PTSD [17]. Although the majority of previous studies focusing on traumatic childbirth and birth-related PTSD have used the DSM-IV criteria, the DSM-IV/DSM-5 comparison study conducted by Kilpatrick and colleagues [18] concluded that 60% of PTSD cases that met the DSM-IV criteria but not the proposed DSM-5 PTSD criteria were excluded from DSM-5. 

Indeed, some significant conceptual changes concerning traumatic events and PTSD need to be mentioned. With DSM-5, some important differences concerning PTSD diagnoses in comparison with the last version, DSM-IV, have been highlighted [19]. According to the definition of the 5th version of the DSM criteria (DSM-5) [17], a traumatic event is no longer characterized by a subjective component as it was with DSM-IV. The definition of a traumatic event according to DSM-5 is the first and mandatory criterion to diagnose a PTSD and consists of being exposed to death, serious injury or sexual violence, whether actual or potential. Consequently, this definition excludes stressful events that do not involve an immediate threat to life or physical injury, such as psychosocial stressors, although a subjective definition was used in DSM-IV, namely a “threat to physical integrity”, which was the origin of many critiques and the idea that the definition of a traumatic event in DSM-IV was too inclusive [19]. 

According to the DSM-5 criteria, a birth-related PTSD diagnosis needs to validate the definition of a traumatic event (criterion A1) and the individual needs to re-experience the traumatic event related to childbirth (criterion B1), present avoidance symptoms (criterion C1), develop alterations in cognition and mood (criterion D1), experience alterations in arousal and hyperactivity (criterion E1) and experience these symptoms for more than one month (criterion F1). The symptoms must also create distress or a functional impairment (criterion G1) and must validate an exclusion criterion regarding medication, substance use or other illnesses that could explain the symptoms (criterion H1) [17]. 

Although the prevalence of birth-related PTSD and its predictors have been studied in many Western countries before COVID-19 pandemic, there are limited data reported in Switzerland [15] so far to be used as comparative data. We initiated a cross-sectional study to estimate the prevalence of traumatic childbirth, birth-related PTSD, post-partum depression and breastfeeding before the pandemic COVID-19 had occurred, but we started the collection of data during the COVID-19 pandemic. Taking account of this unexpected context for our research, we decided to provide an assessment of stress factors directly related to the COVID-19 pandemic and to analyze their associations with traumatic childbirth and birth-related PTSD, in addition to the known risk factors for birth-related PTSD. 

We wanted to assess whether the COVID-19 pandemic (1) increases the risks of traumatic childbirth, childbirth-related PTSD and post-partum depression; and (2) alters the influence of known factors of childbirth-related PTSD. We hypothesized that stress factors related to the COVID-19 pandemic increase the prevalence rates of traumatic childbirth-related PTSD and post-partum depression and that the influence of known factors involved in these clinical context have been modified during the COVID-19 pandemic. The study’s main objective was to assess the prevalence rates of traumatic childbirth and birth-related PTSD in women who delivered during the COVID-19 pandemic. The main secondary objective was first to analyze the already known factors related to COVID-19 associated with traumatic childbirth and birth-related PTSD, while the final secondary objectives were to evaluate whether the presence of birth-related PTSD may be associated with a greater risk of developing comorbid post-partum depression at one month post-partum and to assess the association of breastfeeding practices with birth-related PTSD.

## 2. Materials and Methods

This was a prospective cohort study of all women who delivered at the University Hospitals of Geneva between 25 January 2021 and 10 March 2022 with two follow-up time-points: an early assessment within 3 days of delivery and a visit at one month post-partum. This hospital manages pregnancies and deliveries at high risk of complications and complex psychosocial situations. The eligibility criteria were any woman older than 18 years who delivered at the Geneva University Hospitals’ maternity wards after 29 weeks of amenorrhea (w.a.) and who consented to participate. We conducted a physical interview in their room during maternity to inform them of and to check the inclusion criteria. Women who did not read or speak fluent French were excluded. The recruitment started after childbirth and within 3 days of the maternity stay. 

Once eligible, the participating mothers received an e-mail on their smartphone and completed an online self-report questionnaire within 3 days of their maternity stay and at one month post-partum. During the first assessment, they completed self-report questionnaires for the following variables: (1) age, nationality and current profession; (2) data about delivery routes, whereby maternal or neonatal complications were collected on medical files, support and information provided by healthcare workers during delivery; (3) antenatal depression; (4) peritraumatic reactions around childbirth; (5) psychiatric history and previous psychiatric treatments; (6) questions related to COVID-19 regarding pregnancy and the maternity stay. During the second assessment, at one month post-partum, the mothers completed self-report questionnaires for the following variables: (1) birth-related PTSD; (2) previous traumatic events according to the DSM-5 criteria; (3) questions related to COVID-19 during the post-partum period; (4) breastfeeding; (5) perceived support by the patient’s entourage. They also received a teleconsultation to look for diagnoses of birth-related PTSD and post-partum depression according to DSM-5 criteria. In case of no answer, three automatic reminders were provided by e-mail. 

Depending on the results of the assessment of birth-related PTSD or post-partum depression, a referral to a mental health specialist was proposed, independently of the study.

We describe our outcomes in detail and the tools we used above for each step of the study.

### 2.1. Primary and Secondary Outcomes

At one month post-partum, the participating women were invited to a clinical interview through a teleconsultation conducted by an experimented psychologist or psychiatrist of research team that had each more than 5 years of clinical experience. To validate the criteria for the presence of traumatic childbirth (primary outcome), the clinician checked if criterion A according to DSM-5 for traumatic events corresponded to the last childbirth; then, the other diagnostic criteria for birth-related PTSD according to the DSM-5 criteria were searched (secondary outcome) [17]. The clinician also searched for the presence of PPD according to the criteria for DSM-5 (secondary outcome) [17]. Information regarding stopping or not breastfeeding at one month was collected during this clinical interview (secondary outcome) [8]. The self-report questionnaire included items for the French validation of the PTSD Checklist for DSM-5, assessing the 20 DSM-5 symptoms of PTSD, which we used to evaluate the intensity of birth-related PTSD [20,21].

### 2.2. Known Risk Factors of Birth-Related PTSD

We searched known risk factors of birth-related PTSD during maternity stays, such as antenatal depression [11,14] using the French translation of the Edinburgh Post-Partum Depression Scale (EPDS), a set of 10 screening questions used to identify women who may have post-partum depression, with each answer being scored from 0 to 3 [22,23] and with a total score higher than 11 being interpreted as antenatal depression. The modalities of childbirth (delivery route, instrumented delivery, maternal and neonatal complications) [7,11] and the level of satisfaction with the quality of care, support and medical information provided by healthcare workers (HCW) were evaluated on a 5-point Likert scale [7,11]. The presence of a peritraumatic reaction linked the delivery was assessed using the French version of The Peritraumatic Distress Inventory, a 13-item self-report questionnaire that measures the level of distress experienced by an individual during and shortly after a traumatic event [24,25], and the Peritraumatic Dissociative Experiences Questionnaire, a self-report inventory used to assess dissociation occurring at the time of the trauma [26,27] which is known as a predictive factor of related to childbirth PTSD [28]. For both tools, the assessment was considered as positive if the total score was higher than 15 [24,26]. The participating women were also interviewed about the use of a previous psychotropic treatment, the presence of previous traumatic events (including traumatic childbirth according to DSM-5 and specific types of previous traumatic events, e.g., physical, sexual aggression, severe disease) [7,11]. We tested the perceived social support using the 5-point Likert scale during the post-partum period [29].

### 2.3. Other Variables and Confounding Factors

We tested potential risk factors of birth-related PTSD that are related to the COVID-19 pandemic and that are not known. We provided questions using the 5-point Likert scale concerning access to care for their pregnancy and the newborn after delivery and access to care for physical or mental health, during pregnancy and the post-partum period, concerning the effects of isolation from their relatives, the economic consequences, the use of teleworking, the fear of being infected by SARS-CoV-2, the perception of the restrictions of visits during maternity stay, the presence of infected persons in close contact with delivering women during COVID-19, the impact of the COVID-19 context on the experience of pregnancy and delivery and the received social support. We also assessed for contamination by SARS-CoV-2 during pregnancy. 

### 2.4. Data Use and Recording

All medical and socio-demographic data were recorded using the REDCap electronic data capture tools hosted at the University Hospitals of Geneva, Switzerland [30]. If the participant did not attend the one month consultation, she was considered as a drop-out. In cases of premature exit from the study, before the first month post-partum, the coded data collected were recorded using REDCap and then used for the analysis of the results.

### 2.5. Statistical Analysis

We anticipated that 200 participants would be needed to be able to estimate the prevalence of birth-related PTSD at 4% with a precision of ±2.7% using a 95% confidence interval (95% CI).

The continuous variables were described using the mean ± standard deviation (SD), median and interquartile range; the categorical variables by frequencies and relative proportions. We reported the prevalence of PTSD due to traumatic childbirth or breastfeeding at one month with the 95% CI using the exact binomial method. We assessed the associations between different factors (socio-demographic, clinical and related to COVID-19) with the four outcomes (traumatic childbirth, birth-related PTSD, post-partum depression and breastfeeding) separately by applying four different logistic regression models. For each outcome, we provided univariate analyses then we constructed multivariable models if we had a minimal number of events per variable. We used a stepwise approach by choosing all variables associated with the outcome at *p* < 0.25 in the univariate analyses, then we kept in the final multivariable models all variables that were significantly associated with the outcome (parsimonious) and also variables that showed a confounding effect by changing the coefficient of regression by more than 25% after being stepped out of the model. We verified the adequacy of each model using the Hosmer–Lemeshow test. We provided the pseudo-R2 value to assess the predictive value of the model. We reported the associations between the factors and outcomes using their odds ratios (ORs) and 95% confidence intervals (95% CI). All *p*-values below 0.05 were considered statistically significant. 

## 3. Results

### 3.1. Description of the Sample

We included 265 women between 25 January 2021 and 10 March 2022. A total of 254 (95.8%) women completed the self-report questionnaire within 3 days of their maternal stay and 225 (88.6%) at one month post-partum; 183 (81.3%) women benefited from a clinical interview with a psychologist or a psychiatrist during the second assessment (Figure 1). 

#### 3.1.1. Socio-Demographic Data, Psychiatric and Traumatic History

The socio-demographic data are summarized in Table 1. The women were aged between 21 and 51 y.o (mean = 34.2 y.o). The participating women were mainly active (*n* = 196, 85.2%), and 124 (48.8%) were not Swiss. Most of the participating women were married or in a stable relationship (*n* = 228, 93.4%) (Table 1).

#### 3.1.2. Obstetrical and Neonatal Complications, Perception of the Medical Care

Among the participating women, 138 (75.8%) had delivered by vaginal route, 40 (22.1%) received assisted delivery, 21 (11.5%) had an elective cesarean and 23 (12.6%) had an emergency cesarean (Table 2). Only 6 (3.3%) of the participating women were exposed to neonatal complications, including neonatal hypoxia, prematurity and fetal growth restrictions, while 20 (11.2%) answered that they were exposed to maternal complications in the self-report questionnaire during the maternity stay, such as episiotomy, perineal tears, post-partum hemorrhage, preeclampsia and neurological issues linked with anesthesia (Table 2). 

Immediately after delivery, 252 (99.2%) women were satisfied (including satisfied, quite satisfied and very satisfied) with the quality of support provided by the healthcare workers during their delivery and 254 (100%) were satisfied (including satisfied, quite satisfied and very satisfied) with the quality of information provided by the healthcare workers during their delivery (Table 2). 

#### 3.1.3. Description of Known Factors of Birth-Related PTSD Related to Mental Health

The results for the different tools assessing the mental health state during pregnancy and peripartum are presented in Table 3. The mean EPDS score was 6.7 ± 4.8, and less than one-quarter of the participating women were likely to have antenatal depression (*n* = 55, 21.7%). The mean PDI score was 18.8 ± 7.6, and more than half of the participating women (*n* = 144, 56.9%) obtained a score suggestive of peritraumatic distress predicting the development of birth-related PTSD. The mean PDEQ score was 14.8 ± 6.1, and less than one-third (*n* = 81, 31.9%) obtained a score suggestive of peritraumatic dissociation predicting the development of birth-related PTSD. During the clinical interview, less than one-quarter of participating women (*n* = 35, 21.1%) lived their childbirth as traumatic.

#### 3.1.4. Description of Factors Related to COVID-19 

We have summarized the results of the questions focusing on potential stress factors related to the COVID-19 pandemic in Table 4. During pregnancy, we found that 41 (16.1%) women were contaminated by SARS-CoV-2. The majority of the women recognized a fear of contamination by SARS-CoV-2 during pregnancy for themselves (124, 57.1%) and for their baby intra utero (140, 55.1%), but less so during their maternity stay (41, 16.1%) (Table 4). During the maternity stay, we also found that the great majority (207, 81.5%) appreciated the restriction of visits due to the context of the COVID-19 pandemic (Table 4). Concerning the isolation from their entourage due to the COVID-19 pandemic, we found that 39 (15.4%) women during pregnancy and 34 (15.4%) during the post-partum period did not experience it positively, while only 47 (18.5%) negatively experienced the restriction of visits during the maternity stay (Table 4).

### 3.2. Traumatic Childbirth and Birth-Related PTSD

#### 3.2.1. Descriptive Analysis of Traumatic Childbirth 

The clinical interview at one month post-partum concluded that 35 women (21.1% 95% CI: 15.1–28.1%) experienced a traumatic event according to DSM-5 after checking that the traumatic event was the last childbirth (primary outcome) (Table 5).

#### 3.2.2. Risk Factors of Traumatic Childbirth 

Using a univariate analysis, traumatic childbirth was significantly associated with previous traumatic events (*p* = 0.001), emergency caesarians (*p* = 0.042), maternal complications (*p* = 0.037), antenatal depression (*p* < 0.001), the PDI score (*p* = 0.001) and the PDEQ score (*p* < 0.001) (Table 6). We found no significant association between traumatic childbirth and socio-demographic data, data related to psychiatric history or answers related to COVID-19 (Table 6).

In the multivariable model, traumatic childbirth was significantly and independently associated with antenatal depression as assessed via the EPDS score (*p* < 0.001), peritraumatic dissociation during and after childbirth assessed via the PDEQ score (*p* < 0.001) and also maternal complications (*p* = 0.013), after adjustment for the confounder neonatal complications (Table 7).

#### 3.2.3. Descriptive Analysis of Birth-Related PTSD 

The clinical interview at one month post-partum concluded that 15 women (9.1%, 95% CI: 5.2–14.6%) presented a diagnosis of birth-related PTSD after checking that the traumatic event was the last childbirth (primary outcome) (Table 5). The mean PCL5 score was 10.6 ± 10.8, and 15 women (7.0%) had a score >31 suggesting birth-related PTSD (95% CI, 4.0–11.3%) (Table 8). 

#### 3.2.4. Risk Factors of Birth-Related PTSD 

A diagnosis of birth-related PTSD showed a significant association with exposure to a traumatic event of any type once or more (*p* = 0.001), neonatal complications (*p* = 0.004), antenatal depression as assessed via the EPDS score obtained during the first three days following childbirth (*p* < 0.001) and clinical reactions to traumatic childbirth assessed via the PDI score (*p* = 0.003) and PDEQ score (*p* = 0.001) (Table 9). 

The only answer related to COVID-19 that looked to be significantly associated with birth-related PTSD was the limitation of access to pre-natal care (*p* = 0.004), but we found no significant associations between birth-related PTSD and the other answers related to the COVID-19 context, such as the fear of being infected by SARS-CoV-2 (*p* = 0.639) and isolation due to the COVID-19 context during pregnancy (*p* = 0.723) (Table 10). We did not construct a multivariable model due to the low number of events (Table 9).
ijerph-19-14246-t010_Table 10Table 10Prevalence of PPD at one month post-partum.Variables *n* (%)PPD according to DSM-5
Yes7 (4.3)No157 (95.7)During univariate analyses, we only found a significant association between PPD and the restriction of visits during the maternity stay (*p* = 0.035) (Table 11).
ijerph-19-14246-t011_Table 11Table 11Variables associated with post-partum depression according to DSM-5 (measures of associations, univariate analyses).Variables Odds Ratio95% CI*p*-ValueSocio-demographic data
Current profession (24 missing data)No eventPart-time job/Full-time job/in training
No job/disability status/on prolonged sick leave
Swiss nationality

0.215No1.00-
Yes2.88(0.54–15.27)
Marital status (10 missing data)

0.067Married or in a relationship1.00-
Single5.13(0.89–29.53)
**Previous traumatic events**


Did you experience a traumatic event?

0.093No1.00-
Yes3.74(0.80–17.42)
If yes, were you exposed repeatedly or extremely frequently to traumatic events? 
NoAll post-partum depression had no repeated episodes of traumatic eventsYes
**Delivery characteristics**


Delivery modes

0.510Vaginal delivery1.00--Elective caesarean1.64(0.17–15.50)0.666Emergency caesarean2.81(0.48–16.33)0.250Instrumented delivery

0.697No1.00-
Yes0.65(0.07–5.73)
Neonatal complications

0.168No1.00-
Yes5.03(0.51–50.04)
Maternal complications

0.879No1.00-
Yes1.18(0.14–10.38)
**Mental health variables during pregnancy and immediately post-partum**


Antenatal depression using EPDS

0.144<111.00-
≥113.18(0.67–14.94)
Peritraumatic distress using PDI

0.977<151.00-
≥150.98(0.21–4.52)
Peritraumatic dissociation using PDEQ

0.346<151.00-
≥150.36(0.04–3.04
**Birth-related PTSD**


PCL-5

0.453<311.00-
≥312.33(0.26–21.31)
Post-natal PTSD 

0.093No1.00-
Yes4.43(0.78–25.12)
Birth-related PTSD with depersonalization
NoNo eventYes
Birth-related PTSD with derealization

0.063No1.00-
Yes10.07(0.88–114.60)
**Answers related to COVID-19**


**During pregnancy**


Positive PCR test during pregnancy

0.329No1.00-
Yes2.33(0.43–12.74)
Number of COVID-19 symptoms for infected1.19(0.89–1.57)0.238Were you afraid of being infected by SARS-CoV-2? 

0.139Not at all/A little1.00-
Moderately/A lot/Extremely 0.19(0.02–1.72)
Were you afraid of your baby being infected by SARS-CoV-2? 

0.174Not at all/A little1.00-
Moderately/A lot/Extremely


Were you isolated from your entourage? 

0.505No1.00--Yes and I did not experience it well2.00(0.17–23.18)0.579Yes and I did experience it well2.81(0.50–15.85)0.243Did you modify your work condition and worked at home?

0.606No (I did not experience it well and I did experience it well)1.00-
Yes (I did experience it well and I did not experience it well)1.50(0.32–6.90)
Did you have financial consequences of the COVID-19 pandemic? 

0.083No or yes it has improved 1.00-
Yes, my situation was worsened3.96(0.83–18.78)
Did you have limitation in access to physical care because of the context of the COVID-19 pandemic? 

0.910Not at all or a little1.00-
Moderately/a lot/extremely1.13(0.13–9.91)
Did you have limitation in access to mental care because of the context of the COVID-19 pandemic? 

0.480Not at all or a little1.00-
Moderately/a lot/extremely2.21(0.24–20.04)
Did you have limitation in access to care because of the context of the COVID-19 pandemic? 

0.431Not at all or a little1.00-
Moderately/a lot/extremely2.43(0.27–22.22)
Do you think that context of the COVID-19 pandemic impacted on your pregnancy experience? 

0.419Yes and it was positive1.00--Yes and it was negative1.68(0.14–20.33)0.682No effect0.52(0.06–4.99)0.575**During maternity stay**


Were you afraid of being infected during your maternity stay? 

0.657Not at all/A little1.00-
Moderately/A lot/Extremely1.65(0.18–14.94)
Do you think that the COVID-19 pandemic impacted on your delivery experience? 

0.128No effect1.00--Yes and it was negative6.80(0.64–72.46)0.112Yes and it was positive4.25(0.72–25.07)0.110During you maternity stay, how did you experience the restriction of visits? 

0.035 *I experienced it well and even appreciated it1.00-
I experienced it negatively5.45(1.13–26.23)
* *p* < 0.05; ** *p* < 0.005.


### 3.3. PPD 

The assessment at one month after delivery showed that only 7 (4.3%, 95% CI 1.7–8.6%) of the women suffered from a PPD according to DSM-5 (Table 5). 

### 3.4. Breastfeeding

We found that most of women (192, 86.9%, CI95% 81.7–91.0%) declared that they were still breastfeeding their child at one month post-partum (Table 12).

During the univariate analyses, none of the tested confounding variables were significantly associated with breastfeeding at one month post-partum (Table 13). In the multivariable model, only two variables were independently associated with breastfeeding at one month post-partum: TSTP with derealization (OR = 0.12; 95% CI: 0.02–0.89. *p* = 0.038) and negative experiences with restrictions of visits during the maternity stay (OR = 0.23; 95% CI: 0.08–0.67, *p* = 0.007) (Table 13). 

## 4. Discussion

To the best of our knowledge, this prospective study was the first one to explore the prevalence of birth-related PTSD and traumatic childbirth according to DSM-5 within the context of the COVID-19 pandemic. 

### 4.1. Prevalence of Birth-Related PTSD and Traumatic Childbirth

The strength of our study was to provide a mixed assessment including self-report questionnaires and clinical interviews with the application of the DSM-5 criteria for PTSD. Our results were, thus, comparable with most of the studies that focused on traumatic childbirth and birth-related PTSD using self-report questionnaires; however, they were based on the DSM-IV criteria [9,11,13,14,16]. 

The prevalence of traumatic birth (21.1%) in our study appears to be not so high in comparison with previous studies [5,16]. Ayers concluded that a traumatic experience associated with delivery validated the criterion of a traumatic event at rates of between 20% and 48% [11]. In this study, the traumatic childbirth information was collected using the definition from DSM-IV-TR [1,2], and perhaps it was overestimated. Indeed, there were deep conceptual changes between DSM-IV and DSM-5 concerning PSTD. Criterion A, which defined traumatic events in DSM-IV, was considered too inclusive, generating overestimation for traumatic events [18,19].

We found that 15 (9.1%, 95% CI: 5.2–14.6%) women presented a diagnosis of birth-related PTSD according to the DSM-5 criteria after checking that the traumatic event was the last childbirth. The birth-related PTSD rate at one month appeared to be not so high in comparison with the estimated rates of between 4% for women in the community (i.e., low risk) and 18.5% for women ‘at risk’ (e.g., history of mental illness) in Ayers’ study [11]. However, the authors used an old definition with a high risk of misclassification bias [18,19]. This methodological point may explain the possible underestimation of birth-related PTSD in comparison with our findings. 

A Swiss prospective study in Lausanne concluded that 20.7% of mothers had probable PTSD, regardless of the delivery mode at one month post-partum [15]. Nevertheless, this higher rate may be explained by the fact that birth-related PTSD was defined by the presence of symptoms, while we based our definition on strict criteria for a PTSD diagnosis. Indeed, Schobinger and colleagues assessed probable PTSD cases using the validated 17-item Post-Traumatic Diagnostic Scale-French version (PDS-F) self-report questionnaire [15,31] based on the DSM-IV criteria, and considered that there was a probable case of birth-related PTSD if one re-experiencing symptom was validated (criterion B), with three avoidance symptoms (criterion C) instead of the two in DSM-5 and two arousal symptoms (criterion E) [17,32]. Therefore, there were no mandatory criteria for alterations of cognition and mood disorders (criterion D), delays (criterion F) or functional impairment (criterion G), and no mandatory criteria for the exclusion of the effects of other diseases or toxic substances, although these criteria are necessary to diagnose childbirth-related PTSD according to the DSM-5 [17,32]. 

The other explanation for this relatively low rate of birth-related PTSD in comparison with the previous studies could be selection bias in the participant recruitment. First, bias may have occurred due to the recruitment process, involving a physical interview conducted in the maternity ward to inform the participants and to collect their consent to participate instead of only involving the completion of self-report questionnaires, as was the case in most of the previous studies [9,11,13,14,16]. Women who felt too much distress in the first days following childbirth or at one month post-partum could refuse to participate in the study to avoid speaking about their traumatic experience, knowing that avoidance is one of the main clinical components following traumatic events. Second, the high level of satisfaction towards healthcare workers could suggest that women who were not satisfied did not participate. However, a low level of satisfaction with the support and information provided by the healthcare workers increases the risk of traumatic childbirth and PTSD related to childbirth. These relatively low prevalence rates of both traumatic childbirth and birth-related PTSD were surprising given the COVID-19 context, because we expected worsened perinatal mental health issues and higher rates of birth-related PTSD and traumatic childbirth. The use of questionnaires related to the perceived impact of COVID-19 gave us the possibility to analyze factors directly related to the COVID-19 context. However, although the descriptive analysis showed clear subjective impacts of COVID-19 on the pregnancy, delivery and post-partum experiences for the participants, there was no association between factors related to the COVID-19 context and childbirth trauma, nor between factors related to the COVID-19 context and birth-related PTSD. 

### 4.2. Known Risk Factors of Birth-Related PTSD 

We confirmed some of the results related to known risk factors of birth-related PTSD, such as antenatal depression, a previous traumatic event of any type, neonatal complications, peritraumatic distress and peritraumatic dissociation [9,11,13,14,16], as well as the usefulness of screening these factors during pregnancy. Beyond the pre-birth factors, we expected that the psychiatric history, which was an important part of our sample, would have increased the risk of birth-related PTSD according to the previous study results [11,13], but we found no association. Beyond the peripartum factors, neonatal complications, peritraumatic distress and peritraumatic dissociation related to childbirth increased the risk of birth-related PTSD. More surprisingly, maternal complications and delivery routes were not associated with birth-related PTSD, although these variables are considered important risk factors [9,11,13,14,16]. We could not confirm the previous data showing that the quality of information and support provided by healthcare workers influenced the rates of birth-related PTSD [2].

The early mental health assessment during the maternity stay highlighted the frequent rates of peritraumatic distress (56.9%) and peritraumatic dissociation (31.9%) related to childbirth, with the latter appearing less easy to identify because it needs to be actively researched. Peritraumatic dissociation and peritraumatic distress were both significantly associated with a traumatic childbirth experience and were predictive of birth-related PTSD. The first important lesson from these results concerning birth-related PTSD is the need to raise the awareness of medical staff and midwives involved in the delivery regarding birth-related PTSD, which is less taught in comparison with PPD, and more especially about peripartum factors of birth-related PTSD and the need for prevention strategies. Attention should be paid to the impact of early psychiatric reactions around childbirth more than the objective medical context of the women; these medical complications were not associated with birth-related PTSD in our study. More especially, attention should be paid to peritraumatic dissociation around childbirth, which is more difficult to identify without specific training. However, this clinical reaction is too often ignored and insufficiently taught during higher education. These results remind us that medical staff and midwives would need to know how to identify these symptoms clinically, their psychopathological significance and their management strategies in the context of childbirth. Peritraumatic dissociation around childbirth makes women disconnect from the present moment; for example, they can experience derealization with a very disturbing perception of the environment than can appear as disformed depersonalization, with the impression that they were not themselves and that they were acting during the childbirth and lost control, or they can present with partial or total amnesia around their childbirth. The results concerning the predictive value of the PDEQ score, EPDS score and PDI score for birth-related PTSD encourage the systematic screening of women at risk of developing birth-related PTSD during their maternity stay with the PDEQ score, which is strongly correlated both with traumatic childbirth and birth-related PTSD. The authors of previous studies focusing on peritraumatic dissociation during childbirth recommended also suggested the screening of peritraumatic dissociation to prevent birth-related PTSD [33]. The goal of early screening for women at risk of developing birth-related PTSD would be to propose prevention strategies and early interventions for mental care that could be trauma-focused [34] or based on integrated psychotherapeutic approaches [35,36], with the choice of intervention depending on several clinical factors. 

Our study showed that peritraumatic dissociation around childbirth, as assessed by PDEQ score, presented good correlation with traumatic childbirth assessed by clinical interview. To the best of our knowledge, there are few previous studies focusing on birth-related PTSD [33,34] that have used the PDEQ [26], although peritraumatic dissociation has been well studied in other traumatic contexts and is considered highly predictive of PTSD [37]. Our study confirmed the interest in the use of the PDEQ during the first days following childbirth to screen women at risk of having a traumatic childbirth and developing birth-related PTSD in the future.

Isolation during post-partum was not associated with birth-related PTSD in our study. We also could not confirm that social support was a protective factor for birth experience [29]. However, the data suggest that social support is a key protective factor for PTSD, regardless of the nature of the trauma, and an international population survey study showed that being married was the most important protective factor for PTSD after traumatic event exposure [38]. In this COVID-19 context, the isolation could have been a consequence of the social distancing enforced to protect patients from contamination, meaning it was consequently better accepted by women.

### 4.3. Risk Factors Related to the COVID-19 Pandemic Context

Our study specifically assessed the influence of the COVID-19 pandemic using self-report questionnaires on the main potential consequences on mental health in the context of perinatality. The COVID-19 context was expected to increase the stress load of pregnant women given the public communication concerning the growing evidence for increased risks of major physical health issues and death for women infected by SARS-CoV-2 during pregnancy. Several studies supported pregnancy as a risk factor for severe disease associated with COVID-19. COVID-19 increases the risk for pregnant women to be admitted to an intensive care unit (ICU), the risk of death and the risk of adverse pregnancy outcomes such as preeclampsia, pre-term birth and stillbirth [39], and pregnant women seem to be more susceptible to being infected in comparison with non-pregnant women, but this higher susceptibility towards SARS-CoV-2 remains to be confirmed [40]. For these reasons, the vaccination of pregnant women began to be systematically recommended in most of countries. In Geneva, vaccination began to be proposed for pregnant women after March 2021, while recruitment for our study began 25 January 2021 and ended 10 March 2022. 

Very few of the participants considered there to be an impact of the COVID-19 pandemic context on their pregnancy or their delivery experience. The answers related to the access to prenatal and post-natal care did not show limitations for the great majority of the participants, suggesting the efficient management of care access for perinatality in Geneva, which is not representative of other countries.

Surprisingly, although we could expect that COVID-19 pandemic context would increase the risk of birth-related PTSD due to the potential risk of death in cases of contamination, we did not confirm results found by others who concluded that women giving birth during the pandemic had an increased rate of traumatic childbirth and birth-related PTSD compared with women giving birth before the pandemic [41]. Our study did not provide any comparison group to assess the existence of differences as compared to the pre-pandemic period. Although the authors of this study suggested that this increased rate of traumatic childbirth and birth-related PTSD during the COVID-19 pandemic was explained by a fear of virus exposure for the mother or newborn during their hospital stay, reduced social support because of the restriction of visits and a negative subjective experience of pregnancy and childbirth due to the COVID-19 context, we did not confirm this hypothesis. We only found that the limitation in accessing prenatal care was associated with a higher odds ratio of birth-related PTSD. This result suggested that the women who suffered from limited access to prenatal care were more anxious during their pregnancy and more at risk of developing birth-related PTSD.

### 4.4. Prevalence of Post-Partum Depression at One Month

We confirmed the probable protective factor of the COVID-19 pandemic on PPD that was also found in a previous study in comparison with women assessed before the COVID-19 pandemic [42], but we did not confirm an increased risk of post-partum depression as some authors did [43]. Indeed, we found a very lower rate of 4% of PPD in our study in comparison with studies before COVID-19, concluding that the prevalence of PPD in healthy mothers was 12% [44]. The hypothesis used to explain this protective factor of COVID-19 is that it could be linked with the change to working from home, which could have facilitated the pregnancy by giving a feeling of safety and comfort to the women as they did not need to go out of their home and take certain risks during their pregnancy that they would have taken if it was not for the mandatory working from home arrangement. Additionally, we could argue that the COVID-19 context permitted the presence of the partner at home alongside the mother and reduced feelings of isolation for pregnant women. We did not find an increased risk of PPD for women who developed a birth-related PTSD, although we expected an association according to the previous data [9,11,14]. On the contrary, we found that there was a non-significant positive association between work at home and post-partum depression rate at one month (*p* = 0.6, OR = 1.5(0.32–6.29)). Finally, knowing that the PPD had been assessed in according with DSM-5 criteria, it could look low in comparison with usual PPD rates assessed by EPDS. 

### 4.5. Breastfeeding at One Month

We found that the large majority of the participants (192, 86.9%) maintained breastfeeding at one month post-partum. This result was close to those of a previous survey focusing on breastfeeding. We can explain the high levels of breastfeeding before and during COVID-19 by the promotion of breastfeeding in the maternity wards of the Geneva University Hospitals and by the midwife visits at home until the 56th day post-partum [45].

Breastfeeding was not associated with birth-related PTSD, although previous data showed that most women who suffered from birth-related PTSD did not initiate breastfeeding and stopped before 12 months [8]. Avoidance towards the newborn can be characteristic of birth-related PTSD, which can lead to breastfeeding cessation but also conversely to a fusion relationship with the newborn underpinned by the hypervigilance of the mother. In this second option, breastfeeding is not stopped and a contrario is maintained despite the maternal suffering. However, we found a very low prevalence of post-partum depression, which limited the power of the univariate and multivariate analyses concerning their risk and protective factors. 

We did not find an association between breastfeeding at one month post-partum and other risk factors such as perceived social support, characteristics of delivery, peritraumatic dissociation and peritraumatic distress related to childbirth, neither did we find associations with post-partum depression. The answers related to COVID-19 did not reveal either an influence of the COVID-19 pandemic on breastfeeding. The main hypothesis is a methodological bias due to the fact that we did not collect data regarding mixed breastfeeding. In our study, all women declaring that they maintained breastfeeding were considered as actually maintaining breastfeeding, even if the breastfeeding was mixed and completed with artificial milk. Another explanation is that we assessed breastfeeding at one month given that the previous study highlighted associations between breastfeeding at 12 months and birth-related PTSD [8]. Finally, the breastfeeding assessment should have been completed using a bonding assessment to explore more finely mother–infant problems in cases of traumatic childbirth, birth-related PTSD and post-partum depression. 

### 4.6. Limitations

Our study had several limitations. The main one was the lack of a control group of women screened before COVID-19. We were not able to calculate the prevalence of birth-related PTSD without the COVID-19 pandemic context, so we could not rigorously assess the impact of the COVID-19 context. However, the use of questionnaires related to the perceived impact of COVID-19 gave us the possibility to analyze factors directly related to the COVID-19 context. The other limitation of our study was the lack of an assessment of known risk factors of post-natal PTSD such as pain and previous deliveries or miscarriages, as such data would have been useful. The authors of a recent meta-analysis concluded that pain was associated with traumatic childbirth and birth-related PTSD [16]. We assessed the main pre-birth vulnerability factors, which were antenatal depression and a previous traumatic childbirth, but we did not assess the fear of childbirth and poor health or complications that Ayers highlighted as a risk factor [11]. An investigation of poor health conditions would have been useful regarding the COVID-19 context, because some poor health conditions can also constitute medical risk factors for a severe form of COVID-19 that could increase the risk of birth-related PTSD directly and indirectly by generating maternal medical issues. We excluded the women who did not speak and write fluent French, meaning we could not explore the potential influence of linguistic barriers or migration on birth-related PTSD and traumatic childbirth. Additionally, we did not assess the vaccination status to assess the potential influence on mental health features, and especially on birth-related PTSD and traumatic childbirth, of being vaccinated against SARS-CoV-2. Finally, our study was conducted in a very particular period during the COVID-19 pandemic. However, the study population we followed here was similar to the usual population followed in our maternity ward outside of the COVID-19 pandemic, so we are confident that our findings could be generalized to a broader population, at least to the population usually taken into care in our hospital. The findings related to traumatic events could, however, be higher than outside the context of the pandemic due to the association between these events and the sanitary crisis. We could also extrapolate that in other institutions having the same activities as ours in terms of the case mix, we would have observed similar prevalence rates of traumatic childbirth and child-related PTSD. 

## 5. Conclusions

To the best of our knowledge, this is the first study to have explored traumatic childbirth and birth-related PTSD according to DSM-5 in Switzerland during the COVID-19 pandemic and the related risk and protective factors. The prevalence rates of traumatic childbirth and birth-related PTSD were not as high as expected. The results regarding known risk factors such as antenatal depression, a previous traumatic event of any type, neonatal complications, peritraumatic distress and peritraumatic dissociation were consistent with the literature for birth-related PTSD, but we found no influence of a psychiatric history or of the delivery route. Women who suffered from limited access to pre-natal care due to the COVID-19 pandemic during their pregnancy were more at risk of developing a birth-related PTSD, but we were not able to draw a conclusion regarding the influence of the other factors related to the COVID-19 pandemic, such as fear of contamination, isolation or socio-economic change due to COVID-19, on birth-related PTSD or traumatic childbirth. During the maternity stay, the approach for the prevention of birth-related PTSD should focus on antenatal depression screening, which appears as the main antenatal risk factor, and on peritraumatic dissociation, which appears as the main peripartum factor, so as to detect women at risk of being exposed to traumatic childbirth and to developing birth-related PTSD. Attention should be placed on the education of the medical staff and midwives regarding the post-natal PTSD risks so as to better clinically identify and manage peritraumatic dissociation around childbirth. 

## Figures and Tables

**Figure 1 ijerph-19-14246-f001:**
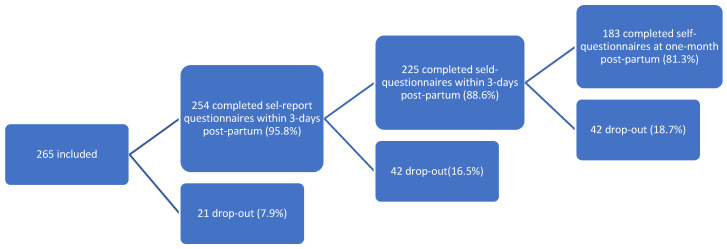
Flowchart of inclusion of women admitted for delivery between 25 January 2021 and 10 March 2022.

**Table 1 ijerph-19-14246-t001:** Socio-demographic data and psychiatric and traumatic event history (*n* = 254).

Variables	
Age (*n* = 247, 8 missing), mean (±SD, median: interquartile range)	34.2(±4.8, 34: 31–37)
Current profession (*n* = 230, 24 missing), *n* (%)	
Part-time job/full-time job/in training	196 (85.2)
No job/disability status/on prolonged sick leave	34 (14.8)
Marital status (*n* = 244, 10 missing), *n* (%)	
Married or in a stable relationship	228 (93.4)
Single	16 (6.6)
Nationality (*n* = 254), *n* (%)	
Swiss	124 (48.8)
Other	130 (51.2)
Psychiatric history (*n* = 254, 34 missing), *n* (%)	
During your life, did you ever see a mental health professional?	
No	91 (41.4)
Yes	129 (58.6)
Were you ever hospitalized in psychiatry?	
No	214 (97.3)
Yes	6 (2.7)
Were you ever treated by psychotropes?	
No	165 (75.0)
Yes	55 (25.0)
Previous traumatic events (*n* = 221, 33 missing), *n* (%)	
Exposure once or more to a traumatic event of any type	
No	162 (73.3)
Yes	59 (26.7)
If yes, nature of traumatic events?	
Physical aggression	16 (7.1)
Sexual aggression	29 (12.9)
Accident	13 (5.8)
Natural disaster	2 (0.90
Attack	1 (0.4)
Sever disease	18 (8.0)
If yes, was it linked to a history traumatic delivery?	
No	19 (83.0)
Yes	10 (17.0)
If yes, was it directly linked with several traumatic events?	
No	19 (32.2)
Yes	40 (67.8)
If yes, were you direct witness to one or several traumatic events to others?	
No	37 (62.7)
Yes	22 (37.3)
If yes, was it linked to one or several traumatic events to your family members or beloved ones?	
No	40 (67.8)
Yes	19 (32.2)
If yes, were you exposed repeatedly or extremely frequently to traumatic events?	
No	48 (81.4)
Yes	11 (18.6)

**Table 2 ijerph-19-14246-t002:** Delivery routes, maternal and neonatal complications and perceptions of support and information provided by healthcare workers.

Variables	
Delivery routes (*n* = 181), *n* (%)	
Vaginal delivery	138 (75.8)
Instrumented delivery	40 (22.1)
Elective cesarean	21 (11.5)
Emergency cesarean	23 (22.6)
Neonatal issues, *n* (%)	
No	174 (96.7)
Yes	6 (3.3)
If yes, reasons	
Neonatal hypoxia	3 (1.6)
Fetal growth restriction	2 (1.0)
Prematurity <37 WA	1 (0.4)
Maternal issues (*n* = 254), *n* (%)	
No	159 (88.8)
Yes	20 (11.2)
If yes, reasons	
Perineal tears	9 (4.9)
Post-partum hemorrhage	4 (2.2)
Preeclampsia	3 (1.6)
Neurological issues secondary to anesthesia	2 (1.1)
Other maternal issues	2 (1.1)
Neonatal issues (*n* = 180), *n* (%)	
No	174 (96.7)
Yes	6 (3.3)
Satisfying perceived support by healthcare workers ^1^ (*n* = 254), *n* (%)	
No	2 (0.8)
Yes	252 (99.2)
Satisfying Perceived information by healthcare workers ^1^ (*n* = 254), *n* (%)	
No	0 (0)
Yes	254 (100.0)

^1^ Considered as satisfied if the woman answered satisfied, quite satisfied or very satisfied versus unsatisfied or not satisfied enough.

**Table 3 ijerph-19-14246-t003:** Mental health state during pregnancy and peripartum (*n* = 254).

Variables	Mean (SD; Median: Interquartile Range)	*n* (%)
Antenatal depression assessed by EPDS score (*n* = 253)		
EPDS (continuous)	6.7 (±4.8, 6: 3–10)	
EPDS score (dichotomous)		
<11 (depression not likely)		198 (78.3)
≥11 (probable depression)		55 (21.7)
Peritraumatic distress around childbirth assessed by PDI (*n* = 254)		
PDI (continuous)	18.8 (±7.6, 16: 14–21)	
PDI score (dichotomous)		
<15		109 (43.1)
≥15		144 (56.9)
Peritraumatic dissociation around childbirth assessed by PDEQ (*n* = 254)		
PDEQ (continuous)	14.8 (±6.1, 13: 10–17)	
PDEQ score (dichotomous)		
<15		173 (68.1)
≥15		81 (31.9)

**Table 4 ijerph-19-14246-t004:** Answers related to stress factors related to the COVID-19 pandemic.

Variables	
**During pregnancy**	
Positive PCR test for SARS-COV-2 during pregnancy, *n* (%)	
No	213 (83.9)
Yes	41 (16.1)
Do you think the fact that someone at home had COVID-19 had a negative impact on the conduct of your pregnancy? *n* (%)	
Not at all	37 (90.2)
A little	2 (4.9
Moderately	0 (0)
A lot	1 (2.4)
Extremely	1 (2.4)
Were you afraid of being infected by SARS-CoV-2? *n* (%)	
Not at all/A little	93 (42.9)
Moderately/A lot/Extremely	124 (57.1)
Were you afraid of your baby being infected by SARS-CoV-2? *n* (%)	
Not at all/A little	114 (44.9)
Moderately/A lot/Extremely	140 (55.1)
Were you isolated from your entourage because of the COVID-19 pandemic? n (%)	
No	127 (50.0)
Yes and I did not experience it well	39 (15.4)
Yes and I did experience it well	88 (34.7)
Did you modify your work conditions and work from home? *n* (%)	
No and I did not experience it well	22 (8.7)
No and I did experience it well	102 (40.3)
Yes and I did not experience it well	13 (5.1)
Yes and I did experience it well	116 (45.9)
Did you have financial consequences of the COVID-19 pandemic? *n* (%)	
Yes, my situation has improved	9 (3.5)
Yes, my situation was worsened	37 (14.6)
No	208 (81.9)
Did you have financial consequences of COVID-19? *n* (%)	
Yes, my situation was worsened	37 (14.6)
No or yes it has improved	217 (85.4)
Did you have limitations in access to mental care due to COVID-19? *n* (%)	
Not at all or a little	233 (91.7)
Moderately/a lot/extremely	21 (8.3)
Did you have limitation in access to prenatal care due to COVID-19? *n* (%)	
Not at all or a little	235 (92.9)
Moderately/a lot/extremely	18 (7.1)
Do you think that COVID-19 pandemic impacted the conduct of your pregnancy? n (%)	
Yes and it was positive	26 (10.2)
Yes and it was negative	45 (17.7)
No effect	183 (72.1)
Do you think that the COVID-19 pandemic context led to pregnancy complications? *n* (%)	
No	244 (96.1)
Yes	10 (3.9)
**During maternity stay**	
Were you afraid of being infected during your maternity stay? *n* (%)	
Not at all/A little	212 (83.8)
Moderately/A lot/Extremely	41 (16.2)
Do you think that the COVID-19 pandemic context impacted your delivery experience? *n* (%)	
Yes and it was positive	10 (3.9)
Yes and it was negative	27 (10.6)
No effect	217 (85.4)
During you maternity stay, how did you experience the restriction of visits? *n* (%)	
I experienced it well and even appreciated it	207 (81.5)
I experienced it negatively	47 (18.5)
**During post-partum**	
Do you think that the context of the COVID-19 pandemic had some effects post-partum? *n* (%)	
Yes and it was positive	33 (15.0)
Yes and it was negative	53 (24.1)
No effect	134 (60.9)
Were you isolated from your entourage due to the context of the COVID-19 pandemic post-partum? *n* (%	
No	141 (63.8)
Yes and I did not experience it well	34 (15.4)
Yes and I did experience it well	46 (20.8)
Did you have limitation in care access post-partum due to the COVID-19 pandemic? *n* (%)	
Not at all	194 (87.8)
A little	18 (8.1)
Moderately	5 (2.3)
A lot	3 (1.4)
Extremely	1 (0.4)
Did you have limitations in physical care access due to the COVID-19 pandemic? *n* (%)	
Not at all/A little	212 (95.9)
Moderately/A lot/Extremely	9 (4.1)
Did you have limitations in mental care access due to the COVID-19 pandemic? *n* (%)	
Not at all/A little	212 (96.4)
Moderately/A lot/Extremely	8 (3.6)
Did you have limitations in your baby’s access to care due to the COVID-19 pandemic? *n* (%)	
Not at all/A little	217 (98.6)
Moderately/A lot/Extremely	3 (1.4)

**Table 5 ijerph-19-14246-t005:** Prevalence of traumatic childbirth defined by criterion A for traumatic events according to DSM-5.

Variables	*n* (%)	95% Confidence Interval
Traumatic childbirth according to DSM-5		
No validation of criterion A	131 (78.9)	
Validation of criterion A	35 (21.1)	15.1–28.1%

**Table 6 ijerph-19-14246-t006:** Variables associated with traumatic childbirth according to the DSM-5 criteria (measures of associations, univariate analyses).

Variables	OR	95% CI	*p*-Value
**Socio-demographic data**			
Current profession			0.689
Part-time job/Full-time job/in training	1.00	-	
No job/disability status/on prolonged sick leave	0.79	(0.25–2.51)	
Swiss nationality			0.865
No	1.00	-	
Yes	0.94	(0.44–1.98)	
Marital status			0.903
Married or in a relationship	1.00	-	
Single	1.09	(0.28–4.19)	
**Psychiatric history**			
During your life, did you ever see a mental health professional?			0.615
No	1.00	-	
Yes	1.22	(0.56–2.63)	
During your life, were you ever hospitalized in psychiatry?			
No	No event, omitted		
Yes			
During your life, were you ever treated by psychotropes?			0.324
No	1.00	-	
Yes	1.51	(0.66–3.44)	
**Previous traumatic events**			
Did you experience traumatic event?			0.001 **
No	1.00	-	
Yes	4.00	(1.82–8.79)	
Physical traumatism			0.010*
No	1.00	-	
Yes	4.39	(1.43–13.53)	
Sexual harassment			0.034 *
No	1.00	-	
Yes	2.91	(1.09–7.82)	
Accident			0.368
No	1.00	-	
Yes	1.94	(0.46–8.17)	
Natural disaster			
No			
Yes	No event		
Assault			
No			
Yes	No event		
Lethal disease			0.047 *
No	1.00	-	
Yes	3.16	(1.02–9.80)	
Previous traumatic childbirth			0.032 *
No	1.00	-	
Yes	4.17	(1.13–15.32)	
**Delivery characteristics**			
Instrumented delivery			0.671
No	1.00	-	
Yes	1.21	(0.51–2.86)	
Delivery modes			0.061
Vaginal delivery	1.00	-	-
Elective caesarean	0.49	(0.11–2.27)	0.362
Emergency caesarean	2.68	(1.04–6.92)	0.042 *
Instrumented delivery			0.671
No	1.00	-	
Yes	1.21	(0.51–2.86)	
Neonatal issues			0.101
No	1.00	-	
Yes	3.97	(0.76–20.59)	
Maternal issues			0.037 *
No	1.00	-	
Yes	2.86	(1.07–7.69)	
**Mental health variables during pregnancy and immediately post-partum**			
Score EPDS			<0.001 **
<11	1.00	-	
≥11	9.18	(3.90–21.60)	
Score PDI			0.001 **
<15	1.00	-	
≥15	4.69	(1.82–12.05)	
Score PDEQ			<0.001 **
<15	1.00	-	
≥15	5.95	(2.67–13.25)	
Quality of the support from healthcare workers during delivery			0.757
Satisfactory	1.00	-	-
Mostly satisfactory	0.86	(0.13–5.55)	0.872
Very satisfactory	1.29	(0.26–6.32)	0.752
**Answers related to COVID-19**			
**During pregnancy**			
Positive test during pregnancy			0.361
No	1.00	-	
Yes	1.57	(0.60–4.12)	
Fear of being infected by SARS-CoV-2			
Not at all/A little	1.00	-	0.840
Moderately/A lot/Extremely	0.92	(0.41–2.05)	
Fear for the baby of being infected by SARS-CoV-2 in utero			0.540
Not at all/A little	1.00	-	
Moderately/A lot/Extremely	1.27	(0.59–2.71)	
Isolation from the entourage because of COVID-19 pandemic			0.928
No	1.00	-	-
Yes and I did not experience it well	0.79	(0.24–2.63)	0.701
Yes and I did experience it well	0.94	(0.42–2.11)	0.887
Did you have limitation in access to care due to the COVID-19 pandemic context?			0.056
Not at all or a little	1.00	-	
Moderately/a lot/extremely	2.57	(0.98–6.76)	
Did you have limitation in access to mental care due to the COVID-19 pandemic context?			0.120
Not at all or a little	1.00	-	
Moderately/a lot/extremely	2.56	(0.78–8.39)	
Did you have limitation in access to prenatal care due to the COVID-19 pandemic context?			0.214
Not at all or a little	1.00	-	
Moderately/a lot/extremely	2.27	(0.62–8.24)	

* *p* < 0.05; ** *p* < 0.005.

**Table 7 ijerph-19-14246-t007:** Variables associated with traumatic childbirth defined by criterion A for traumatic events according to the DSM-5 step-wise approach (*p* < 0.25 for selection and parsimonious multivariate analyses).

Variables	OR	95% CI	*p*-Value
Neonatal complications			0.098
No	1.00	-	
Yes	5.78	(0.72–46.30)	
Maternal complications			0.013 *
No	1.00	-	
Yes	4.68	(1.38–15.91)	
Score PDEQ			<0.001 **
<15	1.00	-	
≥15	5.25	(2.11–13.08)	
Score EPDS			<0.001 **
<11	1.00	-	
≥11	9.67	(3.61–25.91)	

Hosmer–Lemeshow, *p* = 0.7389; pseudo-R2 = 0.2840; * *p* < 0.05; ** *p* < 0.005.

**Table 8 ijerph-19-14246-t008:** Birth-related PTSD at one month post-partum.

Variables	Mean (±SD, Median, Interquartile Range)	*n* (%)
Birth-related PTSD diagnosis ^1^ (*n* = 166)		15 (9.1)
Total score PCL-5 (*n* = 213)		
PCL-5 (continuous)	10.6 (±10.8, 7: 3–15)	
PCL-5 score (dichotomous)		
<31		198 (93.0)
>=31		15 (7.0)

^1^ Diagnosis based on DSM-5 criteria and established during a semi-structured interview conducted by a clinician.

**Table 9 ijerph-19-14246-t009:** Variables associated with a birth-related PTSD diagnosis according to DSM-5 at one month post-partum (measures of associations, univariate analyses).

Variables	Odds Ratio	95% CI	*p*-Value
Socio-demographic data			
Current profession (24 missing data)			0.960
Part-time job/Full-time job/in training	1.00	-	
No job/disability status/on prolonged sick leave	0.96	(0.20–4.60)	
Swiss nationality			0.295
No	1.00	-	
Yes	1.78	(0.60–5.26)	
Marital status (10 missing data)			0.439
Married or in a relationship	1.00	-	
Single	1.89	(0.38–9.45)	
**Psychiatric history**			
Existing previous mental health follow-up			0.249
No	1.00	-	
Yes	2.01	(0.61–6.62)	
Previous psychotropic treatment			0.640
No	1.00	-	
Yes	0.73	0.20–2.73	
Previous hospitalization in psychiatry			
No			
Yes	No event		
**Previous traumatic events**			
Exposure once or more to a traumatic event of any type			0.001 **
No	1.00	-	
Yes	6.51	(2.09–20.34)	
**Delivery characteristics**			
Instrumented delivery			0.325
No	1.00	-	
Yes	1.77	(0.57–5.55)	
Delivery modes			0.320
Vaginal delivery	1.00	-	-
Elective caesarean	0.63	(0.08–5.25)	0. 672
Emergency caesarean	2.40	(0.68–8.44)	0.172
Instrumented delivery			0.325
No	1.00	-	
Yes	1.77	(0.57–5.55)	
Neonatal complications			0.004 **
No	1.00	-	
Yes	12.25	(2.23–67.40)	
Maternal complications			0.895
No	1.00	-	
Yes	1.11	(0.23–5.33)	
**Mental health variables during pregnancy and immediately post-partum**			
Antenatal depression using EPDS			<0.001 **
<11	1.00	-	
≥11	10.58	(3.32–33.72)	
Peritraumatic distress around childbirth using PDI			0.030 *
<15	1.00	-	
≥15	5.39	(1.18–24.72)	
Peritraumatic dissociation around childbirth using PDEQ			0.001 **
<15	1.00	-	
≥15	7.63	(2.30–25.34)	
**Answers related to COVID-19**			
**During pregnancy**			
Positive test during pregnancy			0.358
No	1.00	-	
Yes	0.38	(0.05–3.01)	
Fear of being infected by SARS-CoV-2			0.639
Not at all/A little	1.00	-	
Moderately/A lot/Extremely	0.75	(0.23–2.46)	
Fear for the baby of being infected by SARS-CoV-2 in utero			0.477
Not at all/A little	1.00	-	
Moderately/A lot/Extremely	0.68	(0.23–1.97)	
Isolation from the entourage because of COVID-19			0.723
No	1.00	-	-
Yes and I did not experience it well	1.69	(0.40–7.16)	0.475
Yes and I did experience it well	0.94	(0.28–3.11)	0.919
Change of financial situation because of the COVID-19 context			0.295
No or yes it has improved	1.00	-	
Yes, my situation was worsened	1.92	(0.57–6.55)	
Limitation in access to physical care			0.092
Not at all or a little	1.00	-	
Moderately/a lot/extremely	2.96	(0.84–10.42)	
Limitation in access to mental care			0.860
Not at all or a little	1.00	-	
Moderately/a lot/extremely	0.83	(0.10–6.84)	
Limitation in access to prenatal care			0.004 *
Not at all or a little	1.00	-	
Moderately/a lot/extremely	7.43	(1.88–29.32)	
Do you think that the COVID-19 pandemic context impacted on your pregnancy experience?			0.104
Yes and it was positive	1.00	-	-
Yes and it was negative	1.03	(0.20–5.41)	0.966
No effect	0.31	(0.07–1.32)	0.114
During Maternity stay			
Fear of being infected during your maternity stay			0.597
Not at all/A little	1.00	-	
Moderately/A lot/Extremely	0.57	(0.07–4.61)	
Subjective impact of COVID-19 on the delivery experience?			0.075
No effect	1.00	-	
Yes and it was negative	3.15	(0.89–11.14)	
Subjective impact of the restriction of visits on the maternity stay experience			0.951
I experienced it well and even appreciated it	1.00	-	
I experienced it negatively	0.95	(0.20–4.53)	

Note: * *p* < 0.05; ** *p* < 0.005.

**Table 12 ijerph-19-14246-t012:** Prevalence of breastfeeding at one month post-partum.

Variables	*n* (%)
Are you breastfeeding your child? *N* (%)	
Yes	29 (13.1)
No	192 (86.9)

**Table 13 ijerph-19-14246-t013:** Variables associated with breastfeeding (measures of associations, univariate analyses).

Variables	Odds Ratio	95% CI	*p*-Value
Socio-demographic data			
Current profession (24 missing data)			0.605
Part-time job/Full-time job/in training	1.00	-	
No job/disability status/on prolonged sick leave	1.40		
Swiss nationality			0.991
No	1.00	-	
Yes	0.99	(0.46–2.18)	
Marital status (10 missing data)			0.448
Married or in a relationship	1.00	-	
Single	0.60	(0.16–2.26)	
**Answers related to the COVID-19 pandemic**			
**During pregnancy**			
Positive test during pregnancy			0.738
No	1.00	-	
Yes	1.21	(0.39–3.72)	
Fear of being infected by SARS-CoV-2			0.740
Not at all/A little	1.00	-	
Moderately/A lot/Extremely	1.15	(0.50–2.68)	
Fear of the baby being infected by SARS-CoV-2 in utero			0.285
Not at all/A little	1.00	-	
Moderately/A lot/Extremely	1.53		(0.70–3.37)
Isolation from the entourage			0.292
No	1.00	-	-
Yes and I did not experience it well	4.83	(0.61–38.02)	0.134
Yes and I did experience it well	0.92	(0.41–2.06)	0.834
Change of physical activities because of the COVID-19 context			0.412
Has been maintained as usual	1.00	-	
Has decreased	0.66	(0.25–1.77)	
Change of financial situation because of the COVID-19 context			0.717
No or yes it has improved	1.00	-	
Yes, my situation was worsened	0.82	(0.29–2.34)	
Limitation in access to physical care			0.152
Not at all or a little	1.00	-	
Moderately/a lot/extremely	4.44	(0.58–34.04)	
Limitation in access to mental care			0.553
Not at all or a little	1.00	-	
Moderately/a lot/extremely	1.88	(0.23–15.00)	
Limitation in access to prenatal care	No event		
Not at all or a little			
Moderately/a lot/extremely			
Subjective impact on the pregnancy experience			0.364
Yes and it was positive	1.00	-	-
Yes and it was negative	3.56	(0.59–21.36)	0.166
No effect	1.36	(0.42–4.39)	0.604
**During maternity stay**			
Fear of being infected during the maternity stay			0.291
Not at all/A little	1.00	-	
Moderately/A lot/Extremely	2.24	(0.50–9.95)	
Subjective impact on the delivery experience?			0.710
No effect	1.00	-	-
Yes and it was positive	1.01	(0.12–8.54)	0.996
Yes and it was negative	0.61	(0.19–1.97)	0.409
During you maternity stay, how did you experience the restriction of visits?			0.229
I experienced it well and even appreciated it	1.00	-	
I experienced it negatively	0.56	(0.22–1.44)	
**Traumatic events history**			
Did you experience traumatic event?			0.908
No	1.00	-	
Yes	0.95	(0.40–2.28)	
**Delivery characteristics, neonatal and maternal complications**			
Instrumented delivery			0.932
No	1.00	-	
Yes	1.05	(0.36–3.04)	
Delivery modes			0.170
Vaginal delivery	1.00	-	-
Elective caesarean	0.63	(0.16–2.45)	0.506
Emergency caesarean	0.36	(0.12–1.06)	0.063
Neonatal complications			0.845
No	1.00	-	
Yes	0.80	(0.09–7.20)	
Maternal complications			0.903
No	1.00	-	
Yes	0.92	(0.25–3.43)	
**Mental health predictive factors of birth-related PTSD**			
Antenatal depression using EPDS			0.310
<11	1.00	-	
≥11	0.63	(0.26–1.54)	
PTSD at delivery using PDI			0.568
<15	1.00	-	
≥15	1.26	(0.57–2.75)	
PTSD at delivery using PDEQ			0.967
<15	1.00	-	
≥15	1.02	(0.44–2.37)	
**Birth-related PTSD**			
Birth-related PTSD			0.488
No	1.00	-	
Yes	0.62	(0.16–2.39)	
Birth-related PTSD with depersonalization	No event		
No			
Yes			
Birth-related PTSD with derealization			0.070
No	1.00	-	
Yes	0.16	(0.02–1.16)	
Birth-related PTSD severity assessed by PCL-5	No event		
<31			
≥31			

## Data Availability

The data are available on request due to restrictions for privacy and ethical reasons. The data presented in this study are available on request from the corresponding author. The data are not publicly available due to the collected data being related to mailing addresses.

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
