# Peer review of "Traumatic Childbirth and Birth-Related Post-Traumatic Stress Disorder in the Time of the COVID-19 Pandemic: A Prospective Cohort Study"

_ijerph, 2022, doi:10.3390/ijerph192114246_

Round 1

Reviewer 1 Report

Dear Authors,

Thank you for the opportunity to review this manuscript. The presented research adds essential information to the research on traumatic childbirth and birth-related post-traumatic stress disorder during the COVID-19 pandemic. The strength of this study is in providing mixed assessment, including self-ques- 376 questionnaires, clinical interviews, and application of the DSM-5 criteria of PTSD. This work is well written. It could benefit somewhat if the Authors address some issues mentioned below.

Minor issues: 

  1. Please, pay attention to the minor editorial errors (e.g., p. 3, line 118).
  2. Tables – format following the joural requirements.
  3. p. 27 – remove the technical working instructions for Authors about the Appendices.

Major issues:

  1. Please, state the research questions and the hypotheses explicitly.
  2. Please, provide information on whether the project received a positive opinion from any ethics committee.
  3. "Experimented psychologist?" (p 4, l. 165) - what do you mean?
  4. Please, provide brief descriptions of the measures used in the study, including their psychometric parameters (Method Section)
  5. Please refer more broadly to the issues of generalizability and sample representativeness in your discussion.

Reviewer 2 Report

I want to thank the authors for their interesting study. It has been really well done. Especially with regards to how PTSD was measured, I welcome the rigorous methodology.

There are a number of small issues.

Mainly, some language issue:

«However, childbirth with no objectively medically complied complications, whether obstetric or neonatal, can nevertheless be also experienced as traumatic and lead to complications in the psychic well-being of the mother, the father and and the child,.” – see the end of the sentence. A “,” at the end.

“Consequently, this definition excludes stressful events that do not involve an immediate threat to life or physical injury such as psychosocial stressors although subjective definition was used in DSM-IV such as “threat to physical integrity” that is at the origin of many critics and the idea that the definition of a traumatic event in DSM-IV was too inclusive [19].”

This sentence is strangely structured. There should be a period somewhere. Plus, “at the origin of many critics” – no, at the origin of a lot of critique, not of the critics.

“We initiated a cross-sectional 118 study to estimate the prevalence of traumatic childbirth, birth-related PTSD, post-partum 119 depression and breastfeeding before the pandemic COVID-19 has occurred but we started 120 the collection of data during COVID-19 pandemic.”

Again, make it two sentences.

Then PTSD, PTSS, and post-traumatic stress symptoms.

“Authors of a recent meta-analysis suggested to use the terminology of birth-related post-traumatic stress disorder (PTSD) to describe a PTSD resulting to a traumatic childbirth and concluded that 4.7% of mothers developed a birth-related PTSD and 12.3% of mothers developed birth-related posttraumatic stress (PTSS)[7]”

PTSS=post-traumatic stress symptoms, not “post traumatic stress”.

“Actually, the traumatic childbirth does not necessarily give rise to the development of birth-related PTSD, which is the most serious clinical 64 evolution and can be limited to birth related PTSS that can be post-traumatic stress symptoms that are not sufficient to validate criteria for PTSD diagnosis”. You see here how complicated it gets if you do not use PTSS clearly. So, post-traumatic stress can be post-traumatic stress symptoms… This quickly becomes a semantic discussion. What is post-traumatic stress? Because it is not a diagnosis. Then, what are post-traumatic stress symptoms, if they are different from the post traumatic stress?

I greatly appreciate the fact that the authors took the time to discuss DSM-IV and DSM-V differences in criterion A, because it is something I have complained about a lot when authors do not take it into account.

Another limitation might be that some of the women might be suffering from adjustment disorders. These are ptsd-like symptoms due to non-traumatic stressfull events. I think if the authors would have included this, they would have been able to explain a lot more of the distress and such. Especially with regards to COVID this is something that has been understudied. It also avoids the strict criterion A inclusions of PTSD.  
